# Genome-wide association study for acute otitis media in children identifies *FNDC1* as disease contributing gene

Gijs van Ingen[1,2,*], Jin Li[3,*], André Goedegebure[1], Rahul Pandey[3], Yun Rose Li[3], Michael E. March[3], Vincent W.V. Jaddoe[2,4,5], Marina Bakay[3], Frank D. Mentch[3], Kelly Thomas[3], Zhi Wei[6], Xiao Chang[3], Heather S. Hain[3], André G. Uitterlinden[5,7], Henriette A. Moll[4], Cornelia M. van Duijn[5], Fernando Rivadeneira[5,7], Hein Raat[8], Robert J. Baatenburg de Jong[1], Patrick M. Sleiman[3], Marc P. van der Schroeff[1] & Hakon Hakonarson[3,9,10]

Acute otitis media (AOM) is among the most common pediatric diseases, and the most frequent reason for antibiotic treatment in children. Risk of AOM is dependent on environmental and host factors, as well as a significant genetic component. We identify genome-wide significance at a locus on 6q25.3 (rs2932989, $P_{meta} = 2.15 \times 10^{-09}$), and show that the associated variants are correlated with the methylation status of the *FNDC1* gene (cg05678571, $P = 1.43 \times 10^{-06}$), and further show it is an eQTL for *FNDC1* ($P = 9.3 \times 10^{-05}$). The mouse homologue, Fndc1, is expressed in middle ear tissue and its expression is upregulated upon lipopolysaccharide treatment. In this first GWAS of AOM and the largest OM genetic study to date, we identify the first genome-wide significant locus associated with AOM.

[1] Department of Otolaryngology, Head and Neck Surgery, Erasmus MC, University Medical Center Rotterdam, Rotterdam 3000 CA, The Netherlands. [2] The Generation R Study Group, Erasmus MC, University Medical Center Rotterdam, Rotterdam 3000 CA, The Netherlands. [3] Center for Applied Genomics, Children's Hospital of Philadelphia, Philadelphia, Pennsylvania 19104, USA. [4] Department of Pediatrics, Erasmus MC, University Medical Center Rotterdam, Rotterdam 3000 CA, The Netherlands. [5] Department of Epidemiology, Erasmus MC, University Medical Center Rotterdam, Rotterdam 3000 CA, The Netherlands. [6] Department of Computer Science, New Jersey Institute of Technology, Newark 07102, New Jersey, USA. [7] Department of Internal Medicine, Erasmus MC, University Medical Center Rotterdam, Rotterdam 3000 CA, The Netherlands. [8] Department of Public Health, Erasmus MC, University Medical Center Rotterdam, Rotterdam 3000 CA, The Netherlands. [9] Division of Human Genetics, Children's Hospital of Philadelphia, Philadelphia, Pennsylvania 19104, USA. [10] Department of Pediatrics, The Perelman School of Medicine, University of Pennsylvania, Philadelphia, Pennsylvania 19104, USA. * These authors contributed equally to this work. Correspondence and requests for materials should be addressed to G.v.I. (email: g.vaningen@erasmusmc.nl) or to H.H. (email:hakonarson@chop.edu).

Otitis media (OM) is among the most common pediatric diseases, and the most frequent reason for antibiotic treatment, in children. Interrelated phenotypes include acute OM (AOM), recurrent AOM and OM with effusion. Among OM phenotypes, AOM is defined by the presence of (purulent) fluid in the middle ear accompanied by earache, bulging of the tympanic membrane and fever, usually preceded by an upper respiratory tract infection[1–3], which is the type of OM most frequently encountered by pediatricians, and constitutes the largest number of OM patients. The etiology of OM is one of complex associations between environmental, pathogen, host and genetic risk factors[4]. Heritability is well-established in sibling, twin and family studies, with the fraction of phenotype variability attributed to genetic variation ($h^2$) estimated between 0.22 and 0.74 (refs 5–8).

The genetic susceptibility loci for OM are not well understood. Candidate gene studies, based on biological plausibility of the genes and evidence from model organisms, have examined the association of some relevant genes to OM and yielded significant associations for >20 genes including several interleukin (IL) genes, mucin genes, *TLR4*, *FBXO11*, *TNFα*. (reviewed by Rye *et al.*[9]). However, as with most candidate gene studies, results are conflicting with only a small proportion of the discoveries replicating in independent studies. Recently, agnostic discovery approaches have been adopted to identify the genetic susceptibility loci underlying OM. Single-nucleotide polymorphisms (SNPs) at several loci have been reported to be associated with OM, such as those at 10q26.3, 19q13.43, 17q12, 10q22.3, 2q31.1 for association with chronic and recurrent OM[10–13], 2p23.1 (genes *CAPN14, GALNT14*) and 20q11.21 (*BPIFA* gene) for association with OM in general[14]. None of these signals broke the threshold of genome-wide significance ($P < 5 \times 10^{-8}$), likely due to the small sample size and limited number of genetic variants examined. A recent study of OM via exome sequencing identified the cosegregation of a rare duplication variant in gene *A2ML1* with OM in a Filipino pedigree, and additional *A2ML1* variants were found in otitis-prone American children[15], but these variants were all of very low frequency which could not account for the high prevalence of OM. In addition, there has been no genetic study specifically examining the susceptibility loci for AOM.

We perform genome-wide association studies (GWAS) to identify susceptibility loci for AOM. In the meta-analysis of two discovery cohorts including 825 cases, 7,936 controls and 1,219 cases, 1067 controls respectively, we identify a genome-wide significant locus at on 6q25.3. We subsequently replicate the association in an additional cohort of 293 cases of AOM and 1,719 controls. We further reveal significant correlation between the associated variant and the methylation status and expression levels of the fibronectin type III domain containing 1 (*FNDC1*) gene. Additionally, we demonstrate that the mouse homologue Fndc1 is expressed in the middle ear tissue and is being upregulated under proinflammatory conditions, such as

lipopolysaccharide treatment, suggesting *FNDC1* is the causal gene underlying the association.

## Results

**GWAS identifies AOM susceptibility locus.** We performed a GWAS on AOM with age of onset <3 years old. Both patients and controls were recruited and their genomic DNA (gDNA) was genotyped at the Children's Hospital of Philadelphia (CHOP) and the Generation R Study (Supplementary Figs 1 and 2, Supplementary Tables 1 and 2). In the discovery phase, all samples were genotyped using Illumina HumanHap550-V1/V3 or HumanHap610 or 660-Quad DNA Bead Chips. After quality control filtering, a total of 825 cases and 7,936 controls of European descent retained for analysis at CHOP together with 1219 cases and 1,067 controls at Generation R. Logistic regression analysis under an additive model was performed to assess the association between SNP genotype and AOM at each individual site, including the first three principal components and sex as covariates (Supplementary Fig. 3). The residual genomic inflation factor was 1.00 for analysis at CHOP and 1.01 at the Generation R Study (Supplementary Fig. 4). The site-specific GWAS were subsequently meta-analysed to combine the 460,000 exclusively non-imputed SNPs (minor allele frequency (MAF) >0.01) shared by the two centers, using a sample-size based fixed-effect model. One variant on 6q25.3 with MAF 0.13 in controls (rs2932989, $P_{meta} = 4.36 \times 10^{-08}$) (Table 1, Supplementary Fig. 5) surpassed the genome-wide significance threshold ($P < 5 \times 10^{-08}$) following the meta-analysis. The associated locus was subsequently imputed using the 1,000Genome data set as a reference. A total of 77 genotyped and imputed SNPs in this region showed suggestive association with P values <5 × 10^{-05} (Fig. 1a, Supplementary Table 3).

We subsequently investigated whether the association signal could be replicated in an independent cohort. An additional 293 cases of AOM and 1,719 controls were recruited at CHOP and genotyped for this purpose, using the Illumina HumanOmniExpress-12 v1 chip (Supplementary Table 2). The replication samples were mainly of European and African American ancestry with a minority of subjects of Asian and Hispanic ancestry (Supplementary Fig. 3). The first three principal components and sex were included as covariates in the association analysis. The resulting genomic inflation factor was 1.01 (Supplementary Fig. 4), suggesting population stratification was well controlled. While the top SNP from the discovery cohort, rs2932989 is not directly genotyped on the HumanOmniExpress chip, two proxy SNPs in strong linkage disequilibrium ($r^2 = 1$) with rs2932989 both showed association with AOM ($P < 0.05$) (Supplementary Table 4) in the replication cohort and approached genome-wide significance in the discovery cohort based on imputed genotype data ($P = 7.49 \times 10^{-07}$ for rs578217 and $P = 9.79 \times 10^{-07}$ for rs419009, Supplementary Table 3). Direct interrogation of rs2932989 following imputation of the replication cohort

### Table 1 | Genome-wide significant association of 6q25.3 with acute otitis media.

| SNP | Chr | Pos (hg19) | Gene | A1/A2 | MAF cases/controls | Stage | OR$_{CHOP}$ (95% CI) | P$_{CHOP}$ | OR$_{GenR}$ (95% CI) | P$_{GenR}$ | P$_{meta}$ |
|---|---|---|---|---|---|---|---|---|---|---|---|
| rs2932989 | 6 | 159699284 | *FNDC1* | T/G | 0.17/0.13 | Discovery | 1.41 (1.23, 1.62) | 1.46e$^{-06}$ | 1.25 (1.05, 1.48) | 1.02e$^{-02}$ | 4.36e$^{-08}$* |
| | | | | | | Replication | 1.35 (1.05, 1.73) | 1.55e$^{-02}$ | | | |
| | | | | | | Combined | | | | | 2.15e$^{-09}$† |

A1, minor allele; A2, major allele; Chr, chromosome; CI, confidence interval; MAF, minor allele frequency; OR, odds ratio; P, P value; Pos, position; SNP, single-nucleotide polymorphism.
*P value of meta-analysis at discovery stage.
†P value of meta-analysis of all cohorts.

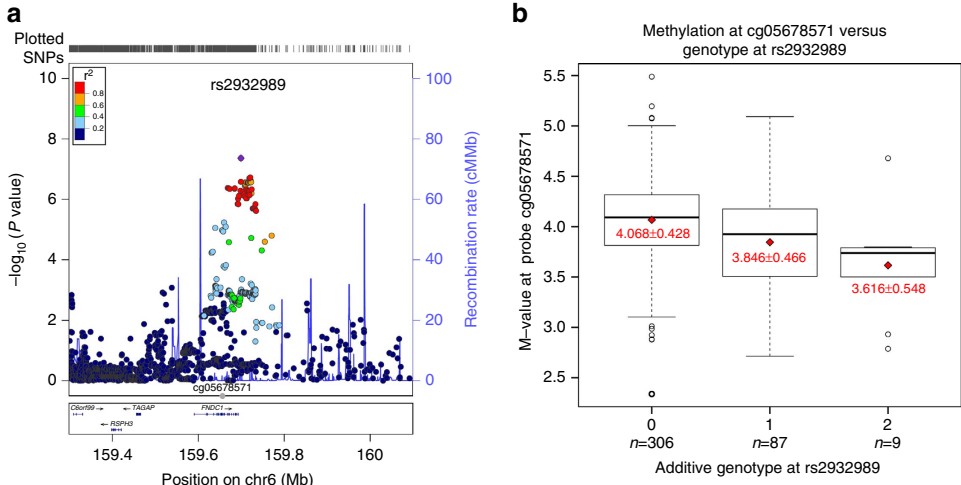

**Figure 1 | The association statistics of locus 6q25.3 with acute otitis media and correlation with methylation status in gene *FNDC1*. (a)** The regional association plot at locus 6q25.3. The SNP chromosomal location on genome build hg19 is indicated on the *x* axis and the negative log10 of *P* value for each SNP is plotted on the left-hand *y* axis. Association statistics were from the meta-analysis of the two cohorts in the discovery phase. The most associated SNP (rs2932989) is shown as purple dot and the other SNPs including both genotyped SNPs and imputed SNPs are coloured according to their linkage disequilibrium with SNP rs2932989. The recombination rates are shown as blue lines. The plot was generated with software LocusZoom[70]. The position of the methylation probe cg05678571 is indicated by a grey dot. (**b**) M-values for methylation probe cg05678571 are plotted against the additive genotype at SNP rs2932989. Dark horizontal lines in the boxplots represent the median of the group, the boxes represent the 25–75% quantiles, and the whiskers of the boxplot extend beyond those quartiles to 1.5 times the interquartile range. Data outside those ranges are represented by points (open circles). Red diamonds indicate the means of each group, and the red text is the mean ± s.d. of each group. The number of individuals with each additive genotype of minor allele T is indicated below the *x* axis.

confirmed the association $P = 0.0155$ (Table 1). Further meta-analysis of the discovery cohorts and the replication cohort yielded a *P* value of $2.15 \times 10^{-09}$ for top SNP rs2932989. We, therefore, demonstrate the first genome-wide significant association of common variation with susceptibility to AOM and replicate the finding in an independent pediatric AOM cohort.

**Test statistics for suggestive loci and candidate genes**. In addition to the genome-wide significant association at 6q25.3 locus, we also observed suggestive evidence for association at eight additional loci ($5 \times 10^{-8} < P$ value $< 5 \times 10^{-5}$; Supplementary Fig. 6 and Supplementary Table 5), among which, *KIF21B, CACNA1S, CRHR2, BDKRB2* and *TPM4* have been reported to be involved in pathogenesis of autoimmune disease, inflammatory disease or response to bacterial infection[16–22]. Interrogating our GWAS data set for candidate genes that have been previously proposed to be involved in pathogenesis of OM, we found 45 out of 82 genes demonstrated evidence of nominally significant association, with intragenic SNPs or nearby SNPs of *P* value < 0.05; Supplementary Table 6), such as *SMAD2, SMAD4, NELL1. BMP5, GALNT13* which are involved in transforming growth factor (TGF)-beta signalling, pro-inflammatory cytokine *IL1B* and its inhibitor *IL1RN*, receptor for fibroblast growth factor (*FGFR1*). TGF-beta, IL-1, fibroblast growth factor (FGF) all have been reported to regulate FNDC1 expression[23,24].

**FNDC1 is expressed in mouse middle ear tissue**. The associated variants at 6q25.3 locus map to an linkage disequilibrium block that contains one gene, the *FNDC1* (Fig. 1). Expression data from both mouse and human tissues (http://www.informatics.jax.org; http://www.eurexpress.org; GTEx database[25]) indicate FNDC1 is expressed in different tissue types with high levels in mesenchymal stem cells, synovial membrane, fetal cartilage, thyroid gland, adipose tissue and others (Supplementary Fig. 7; Supplementary Data 1). In addition, *Fndc1* expression was also found in pharyngeal tissues including the oral epithelium, next to

esophagus tissue, lips, palate and salivary glands (Supplementary Table 7). Database 'Bgee: Gene Expression Evolution' (http://bgee.unil.ch/)[26] included three studies detecting the expression of *Fndc1* in mice ears and one study showed its expression in inner ears[27–29]; additional data from the Shared Harvard Inner-Ear Laboratory database (https://shield.hms.harvard.edu) indicate Fndc1 expression is present in both the cochlea and utricle. We further conducted experiments to examine the expression of Fndc1 in mouse middle ears. By western blotting, we observed not only the expression of Fndc1 in C57/BL6 mice middle ears but also the upregulation of Fndc1 upon lipopolysaccharide treatment compared with control treated with PBS alone (Supplementary Fig. 8).

**AOM variants correlate with *FNDC1* expression and methylation**. In addition to the expression of *FNDC1* in tissues of relevance to the phenotype, data from the GTEx database[25] also indicates the AOM associated SNP, rs2932989, is an expression quantitative trait locus (eQTL) for *FNDC1* in esophageal smooth muscle (Supplementary Fig. 9) ($P = 9.3 \times 10^{-05}$).

Evidence of an association between the AOM-associated variants and *FNDC1* expression was further strengthened following the analysis of methylation data. We found significant association between methylation probe cg05678571 (chr6: $159,660,813 - 159,660,813$, Fig. 1a) and rs2932989 (beta $= -0.200$, $P = 1.43 \times 10^{-6}$, Fig. 1b) and even a stronger association between cg05678571 and SNP rs2501175 (beta $= -0.31$, $P = 7.27 \times 10^{-15}$; rs2501175 meta-analysis $P = 2.89 \times 10^{-07}$) (Supplementary Fig. 10 and Supplementary Table 3). The genomic distance between cg05678571 and rs2932989 is 38.5 kb and that between cg05678571 and rs2501175 is 53.6 kb, both of which are < 1 Mb, the average size of topologically associating domains and gene co-expression clusters in human[30]. Checking the 3D Genome Browser (http://www.3dgenome.org)[31–33], we found that the above two GWAS SNPs and the methylation probe are indeed located in the same topologically associating domain

(Supplementary Fig. 11), suggesting plausible interactions between these loci.

As expected[34], the eQTL and methylation data were inversely correlated. The minor allele (T) at rs2932989 was positively correlated with the *FNDC1* expression levels (Supplementary Fig. 9) and negatively correlated with methylation status of probe cg05678571 in *FNDC1* (Fig. 1b) indicating that methylation of CpG islands in *FNDC1* results in reduced expression. Finally, we used Haploreg[35] to determine if any of the 77 AOM-associated SNPs at the locus overlaps with protein binding domains. Two of the SNPs were found to have CTCF and EGR1 bound based on the ENCODE data[36] (Supplementary Table 8). The same binding motifs were also predicted at the location of the rs2932989-associated methylation probe (cg05678571 CTCF at chr6:159660813-159660831 and EGR1 at chr6:159660869-159660882) suggesting a role for these proteins in the modulation of *FNDC1* expression.

## Discussion

In summary, we have performed the first GWAS of AOM and the largest genetic study of OM to date, the most common medical problem encountered by general pediatric practitioners. We discovered genome-wide significance at a locus on 6q25.3 that contains the *FNDC1* gene and replicated the association in an independent pediatric cohort. We further show that the mouse homologue of this gene is expressed in the middle ear and is upregulated upon lipopolysaccharide treatment. In addition to the tissue expression patterns, the eQTL and methylation data also implicate *FNDC1* as the causal gene underlying the association.

*FNDC1* gene encodes a fibronectin type III domain containing protein. Fibronectin is an important type of extracellular matrix proteins, which interacts with integrins and is involved in cell adhesion, migration, proliferation and differentiation[37]. Only a few studies have been reported regarding the function of FNDC1, which suggest that FNDC1 is involved in multiple cellular processes. *FNDC1* was expressed in heart tissue, with the role of activating G-protein signalling by interacting with Gβγ[38]. This interaction also affects CX43 function and cell permeability, thus sensitizing cells to hypoxic stress in rat neonatal cardiomyocytes and playing an essential role in hypoxia induced apoptosis[39].

While the biological function of FNDC1 has not been well studied, it has been shown that FNDC1 has a role in inflammation. *FNDC1* was first identified as a differentially expressed mRNA from human dermal fibroblasts. Its expression level is increased following treatment with TGF-β, IL-1 and TNF-α[23]. Partially unfolded type III fibronectin module has been shown to induce the expression of IL-8 and TNF-α via activation of the NF-κB signalling pathway, suggesting that fibronectin matrix remodelling may impact cytokine expression[40,41]. Both tissue remodelling and increased levels of cytokines are involved in AOM[42,43]. In line with these observations, in our study, the minor allele (T allele) of SNP rs2392989 confers a higher risk of AOM and is correlated with a lower level of methylation at cg05678571 in *FNDC1*, as well as higher expression of FNDC1. In addition, Fndc1 expression in mouse middle ear was upregulated upon lipopolysaccharide treatment, which is known to be a potent inducer of inflammation, stimulating TGF-β, TNF-α and IL-1 signalling[44–46]. Consistent with the association of *FNDC1* with OM and its expression being modulated by lipopolysaccharide, several important players involved in these signalling pathways of inflammatory responses also showed at least nominal association with OM in our GWAS. *FNDC1* is also one of the differentially expressed genes in primary central nervous system lymphoma compared with normal lymph node

tissues[47], with possible involvement in lymphocyte production. Therefore, the cumulative evidence from literature and the results from our GWAS and mouse experiments suggest that FNDC1 may have a role in the pathogenesis of OM, likely through altered immune or inflammatory response.

The diverse signalling processes that FNDC1 is involved in are inter-related. In an environment of inflammation, hypoxia is typical, with decreased level of oxygen and glucose, as well as increased level of inflammatory cytokines[48]. Hypoxia and hypoxia-inducible factor have an important role in chronic OM and myringotomy reduces hypoxia and inflammation, which are demonstrated by mouse model[49,50]. Hypoxia-inducible factor is of well-known function in hypoxia induced apoptosis[51]. Hypoxia signalling is linked to innate immunity, adaptive immunity and infections. Genes functioning in hypoxia signalling interact with players of NF-κB pathway. (reviewed by Eltzschig and Carmeliet[52]) G protein signalling is of critical roles in various immune functions[53]. The FNDC1 interactor Gβγ has important functions in immune response. For example, it activates PI3Kgamma, regulating inflammation through neutrophil recruitment[54,55]. It also activates RhoGef, involved in lymphocyte chemotaxis and actin polymerization[56]. It would be interesting to examine whether FNDC1 is involved in AOM pathogenesis through any of these signalling pathways.

Among the genes that have been previously proposed to be involved in OM pathogenesis, we found more than half of them exhibited nominally significant association, suggesting consistency between genetic studies and/or with results from model organism studies. It is not surprising that some of the candidate genes did not show significant association, considering the following possible reasons. In our GWAS, we focused on a more defined phenotype of early-onset AOM, which is different from the phenotypes examined in many of the previous OM genetic studies including OM in general, chronic OM or OM with effusion. In addition, different ethnicity of the study population is another influential factor to consider. Furthermore, polymorphisms of candidate genes, proposed based on evidence from differential gene expression, molecular, cellular functions and rodent model studies, may not always present an effect large enough to be captured by GWAS.

The strength of the study lies in the largest sample size and dense genome-wide SNP coverage. Furthermore, we have balanced power and optimal quality control. Compared with previous efforts, this study applied more stringent quality control and more robust correction for population substructure. Indeed, by performing principal-component analysis (PCA) to identify participants of European ancestry, followed by a second PCA on this selected population, we were able to more thoroughly correct for population stratification. By a well-defined phenotype in both populations we were able to combine data from two studies of different designs. Limitation herein is severity of disease, as in one population cases consisted of children who were specifically diagnosed with AOM, likely the more severe episodes, whereas the other population used extensive questionnaires, discovering even the mildest of episodes. These differences were resolved by the increased power of combining the studies, underlined by results showing the same direction of effect and similar effect size between both. Furthermore, the *P* value of homogeneity test in our meta-analysis is >0.9 for the genome-wide significant SNP rs2932989, which indicates that there is no suggestion of between-study heterogeneity.

This study provides evidence of a genetic component influencing susceptibility to AOM in children. By gaining better understanding of the complex polygenetic pathogenesis of AOM, we hope to be able to develop more specific therapies and to ultimately learn which children are most susceptible to this

disease, where earlier clinical intervention may have the most impact.

## Methods

**Statement of ethics.** This study was conducted at the Center for Applied Genomics (CAG) at the CHOP, and in the Generation R Study (GenR) at Erasmus University Medical Center, The Netherlands. The study was approved by the Institutional Review Board at the CHOP, and the Medical and Ethical Review Committee of the Erasmus University Medical Center. Written informed consent was obtained from all participants.

**CHOP population and phenotype definition.** All children in the CAG biobank recruited from the Greater Philadelphia area with both phenotypic and genotypic data available were eligible for this study. Case-status for AOM was defined using ICD-9 codes (Supplementary Fig. 1). Exclusion criteria were established, defined by specific diseases with strong correlation to AOM due to specific defects in anatomy and/or strongly related to susceptibility to upper respiratory tract infections (Supplementary Table 1). Subjects in the CAG database with no history of middle ear disease were labelled as controls (Supplementary Table 2). DNA samples at CHOP were obtained via whole blood venipuncture or saliva samples collected at CHOP and associated health centers in the Philadelphia metropolitan area.

**The Generation R Study population and phenotype definition.** The Generation R Study is a population-based prospective cohort study from fetal life until adulthood[57]. We defined cases and controls using survey data on OM, fever with earache, otorrhea, use of eardrops per subscription by family practitioner or ear, nose and throat (ENT) surgeon. Questionnaires were sent at the ages of 2, 6, 12, 24 and 36 months. As such, case status was established for the phenotype of AOM up to the age of three years, and control status (Supplementary Fig. 2, and Supplementary Table 2). Only children of European ancestry were considered, and exclusion criteria were applied (Supplementary Table 1). At Generation R, whole-blood samples were collected postpartum from the umbilical cord, or at 5 years from venipuncture.

**Discovery phase genotyping and quality control.** gDNA, extracted following the phenol-chloroform protocol, was genotyped using Illumina HumanHap550-V1/V3 or HumanHap610 or 660-Quad DNA Bead Chips at CHOP and the Generation R Study. Quality control was performed through PLINK (software release v1.07; http://pngu.mgh.harvard.edu/purcell/plink/)[58]. Samples with a call rate below 98% at CHOP and below 97.5% at Generation R, or, at either center, with ambiguous sex detected by PLINK or excess of autosomal heterozygosity ($F <$ mean-4 s.d.), were excluded. Only SNPs that were common to all chip types were included in analysis. SNPs with call rate < 95% were excluded, as well as SNPs with MAF < 0.01, and Hardy–Weinberg equilibrium $P$ values < $10^{-5}$. PLINK was used for PCA at the Generation R Study, whereas, at CHOP, PCA was conducted using software Eigenstrat[59]. We excluded subjects that were not of European ancestry and repeated the PCA in the remaining samples to derive the principal components to use as covariates for the association analysis as described below. Genome-wide identity-by-descent analysis was performed using PLINK. Duplicated or cryptically related samples were detected as PI_HAT $\geqslant$ 0.1875; and one from each pair was excluded from further analysis.

**Association analysis and meta-analysis.** After quality control filtering, a total of 825 cases, 7,936 controls and 507,300 SNPs were left for association analyses at CHOP. At Generation R, 469,664 SNPs were available after quality control filtering, a total of 1,219 cases, 1,067 controls. Association analysis was performed at the two sites separately. At both sites, logistic regression analysis under the additive model was performed to assess the association between SNP genotype and AOM. Principal components (PC) were implemented in our model: AOM = SNP + PC1 + PC2 + PC3 + sex. Association analyses were carried out via PLINK at CHOP, and using GRIMP[60,61] at the Generation R Study. We completed a GWAS at each center and meta-analysed SNPs shared by the two centers. As such, ~460,000 genotyped SNPs were available for meta-analysis. We performed sample-sized weighted meta-analyses using METAL software package[62].

**Replication analysis.** To increase power, we included all AOM cases that passed the phenotype selection criteria and were genotyped on the HumanOmniExpress-12v1 array to form the replication cohort. Similarly controls were genotyped on the same array type and had no history of middle ear disease. Samples underwent the same quality control as the discovery cohorts. Logistic regression analysis was conducted including the first three PCs and sex as covariates.

**Regional imputation.** The CHOP discovery and replication cohorts were prephased using SHAPEIT[63,64] version 2, followed by genotype imputation of chromosome 6 using IMPUTE 2 (refs 65,66) against the 1,000 Genome Phase 3 reference (https://mathgen.stats.ox.ac.uk/impute/1000GP%20Phase%203%20haplotypes%206%20October%202014.html). Association testing of the imputed genotypes was carried out in SNPTEST[65] V2. using the missing data likelihood score test, including sex and the first three principal components as covariates. We excluded SNPs with info score < 0.8 or with Hardy–Weinberg equilibrium-test $P$ value < $1 \times 10^{-06}$.

**Analysis of methylation data.** Genome-wide methylation profiles were generated on a total of 854 subjects recruited by the CAG on the Infinium HumanMethylation450 BeadChip Kit according to the manufacturers' protocols. gDNA was isolated from peripheral blood mononuclear cells. Methylation data were exported from the Illumina GenomeStudio, as GenomeStudio output text files containing probe level summarized information. The GenomeStudio output files were loaded into the R statistical package (r-project.org) using the lumi package[67,68]. The lumi package was used for preprocessing of data, which involved quantile colour balance adjustment and background level correction and simple scaling normalization (ssn). M-value density and CpG-site intensity were plotted after normalization, and two aberrant chips were removed. Principle component analysis identified 425 subjects of European ancestry, 374 African Americans, 20 East Asians, and 24 Hispanics among these subjects. Methylation probes known to overlap with common SNPs, were identified and removed using the IMA R package[69]. M-values (the log2 ratio between the methylated and unmethylated probe intensities) were extracted and stored as a matrix. Additive genotypes at rs2932989 for subjects of European ancestry were extracted from existing genotyping data using PLINK. There are a total of 402 subjects of European ancestry without missing genotype at rs2932989 and extreme outlier values of methylation M-values ($\geqslant$ median M-value of the genotype group $\pm$ 3 s.d.). Methylation data in gene *FNDC1* were analysed as the response variable in a linear regression, with genotype at rs2932989 as the predictor variable among these 402 subjects. Sex, age, and 10 genotype-derived principle components were included as covariates. Linear regression and generation of boxplots was performed using base packages in R.

**Animals.** The Institutional Animal Care and Use Committee of the CHOP approved all animal studies. C57/BL6 female mice were used in the study. Mice were randomly distributed into two groups and deeply anesthetized, using ketamine (100 mg per kg) and xylazine (10 mg per kg) via intraperitoneal injection. The experimental group received lipopolysaccharide (2 mg per ml) in PBS, 20 ul per ear derived from Salmonella typhimurium (Sigma-Aldrich), and 0.01 M PBS was injected into the middle ears of the control group. Mice were killed at third day after injection of lipopolysaccharide or PBS.

**Western blot analysis.** For western blot analysis, mice middle ear tissue, including the epithelial lining was removed from animals after decapitation and sonicated in ice-cold NP40 lysis buffer (Invitrogen) containing protease inhibitor (Calbiochem). Proteins extracts were boiled for 5 min in SDS– polyacrylamide gel electrophoresis sample buffer (0.125 M Tris-HCl, 20% glycerol, 4% SDS, and 0.005% bromo-phenol blue) for denaturation. A 20-μg protein sample was separated on a 4–12% NuPAGE Bis-Tris gel (Novex), transferred overnight to a nitrocellulose membrane (Invitrogen), and blocked overnight with 3% BSA tris-buffered saline with Tween (10 mM Tris-HCl (pH 7.5), 150 mM NaCl, and 0.05% Tween 20). The membrane was cut in half with the upper half of the membrane incubated with rabbit anti-FNDC1 polyclonal antibody (H-65; sc-382009) at 1:1000 dilution and the lower half of the membrane probed with rabbit anti-β-Actin polyclonal antibody (N-21; sc-130656) at 1:1000 dilution at room temperature for one hour. Subsequently, the membranes were washed for three times and incubated in a 1:10,000 dilution of horseradish peroxidase– conjugated anti-rabbit secondary antibody (Promega) for 1 h at room temperature and washed again three times. WesternBright enhanced chemiluminescence (ECL) detection system (Advansta) was used to detect bound antibody. Then the band intensities were measured with Image J software (NIH Shareware).

**Data availability.** The data are available from the corresponding authors upon request.

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

## Acknowledgements

We thank all staff at the genotyping center and biorepository at the Center for Applied Genomics at CHOP, and Anis Abuseiris, Karol Estrada, Dr Tobias A. Knoch and Rob de Graaf as well as their institutions Biophysical Genomics, Erasmus MC Rotterdam, The Netherlands, and especially the national German MediGRID and Services@MediGRID part of the German D-Grid, both funded by the German Bundesministerium fuer Forschung und Technology under grants #01 AK 803 A-H and # 01 IG 07015 G for access to their grid resources. Funding for this study was provided by a gift from the Kubert Estate family; an Institutional Development Fund to The Center for Applied Genomics and U01HG006830 from the NHGRI (eMERGE network), as well as funding from the EU 7th Framework Programme under grant agreement number 247642, GEoCoDE.

## Author contributions

H.H. and M.S. were the principal investigators for this study, who conceived this study and supervised its design. Financial support was obtained by H.H. and M.S., in collaboration with R.B. and V.J. G.v.I and J.L. assisted in study design, defined phenotypes, and collected and organized data, with assistance of P.M.S., A.G., F.M. and H.M. Genotyping and sequencing were performed by—and under supervision of—A.U., C.D., K.T. and F.R. Methylation sequencing was supervised by M.M. Mouse experiments were conducted by R.P., M.B. and H.S.H. Analyses were performed by G.v.I and J.L., with assistance of Y.R.L., Z.W. and X.C., who interpreted results in close collaboration with H.H. and M.S. The manuscript was drafted by G.v.I and J.L. with critical input from P.M.S., H.H. and M.S. The manuscript was critically reviewed and approved by all authors.

## Additional information

**Competing financial interests:** The authors declare no competing financial interests.

