## [Peer review file · Nature Communications]

Reviewers' Comments:

Reviewer #1 (Remarks to the Author)

The article reports on FNDC1 as a novel GWAS locus for acute otitis media (AOM), which would be the first locus associated with AOM based on genome-wide statistical significance. The study results are moderately supported by established GWAS methods, including replication and meta-analysis, as well as methylation profile. The results are of particular interest to those who are involved in the genetics of otitis media, for which the previously identified association loci for common variants did not reach the genome-wide statistical significance threshold. There is some concern that there are major differences in phenotyping of each cohort, although if these phenotypic differences result in misclassification bias the expected effect on the study would be to diminish the OR, which might provide some relief that the association signal is robust to misclassification.

What is the p-value for comparison of methylation profiles by additive genotype at rs2932989 (Fig. 1B)?

The authors themselves note that the lack of evidence of FNDC1 expression within the middle ear mucosa or the Eustachian tube is a limitation. Would it be possible for the authors to quickly test middle ear mucosa from a rodent model for FNDC1 expression?

It should be noted that FNDC1 is quite close to the PLG gene, and plg-deficient mice develop chronic otitis media (Eriksson et al. 2006). Is there any evidence that FNDC1 is related to other candidate genes for otitis media by gene ontology?

The authors should also include a list of suggestive loci identified from this study (if any) as these may also be of interest to other researchers.

Reviewer #2 (Remarks to the Author)

This paper by van Ingen et al. reports findings from a replicated GWAS of acute otitis media. The authors identify one locus (6q25.3) with genome wide significance and show that potentially causal SNPs are associated with eQTL and methylation status. This is a high-quality study performed using sound and appropriate statistical analyses. I have made several suggestions for the authors that may help clarify certain aspects of the methodology (points 1 - 8 below).

(1) Results. "We found significant association between methylation probe cg05678571 and rs2932989 ($\beta=0.200$, $P=1.43 \times 10^{-6}$, Fig. 1b) and even a stronger association between cg05678571 and SNP rs2501175 ($\beta=0.31$, $P=7.27 \times 10^{-15}$; rs2501175 meta-analysis $P=2.9 \times 10^{-07}$ ". It would be helpful here in the Results section to indicate the genome distance between cg05678571 and rs2932989, as well as the distance between cg05678571 and rs2501175.

(2) Results. The authors list a number of tissues in which FNDC1 is expressed, including oral epithelium, esophagus tissue, lips, palate, salivary glands, inner ear, cochlea and utricle. The databases referenced by the authors, and other databases, however, include a number of tissue types; it would therefore be helpful to provide a ranked list of tissues and cell types based on FNDC1 expression (highest to lowest FNDC1 expression). For instance, a brief look at GTEx seemed to suggest that FNDC1 expression is highest in thyroid, sigmoid colon, and tibial nerve (not mentioned in the Results section). The BioGPS database indicates that FNDC1 expression is highest in mesenchymal stem cells, synovial membrane and fetal cartilage. These tissue types are not mentioned in the Results section.

(3) Results. "In addition to the expression of FNDC1 in tissues of relevance to the phenotype, data from the GTEx database¹⁶ also indicates the AOM associated SNP, rs2932989, is an eQTL for FNDC1 in esophageal smooth muscle (Supplementary Fig. 6) ($P=1.9 \times 10^{-4}$)."

The authors should indicate whether this p-value reaches the genome-wide significance threshold, given the number of SNP-eQTL pairs evaluated.

(4) Results. "Two of the SNPs were found to have CTCF and EGR1 bound based on the ENCODE data¹⁹ (Supplementary Table 6)." Is the alignment of CTCF and EGR1 binding such that the sites are disrupted by genetic variation at the SNPs? It would be helpful here to indicate the cell types used for the ChIP assays.

(5) Methods. "Methylation status was ascertained for a subset of subjects recruited by the Center for Applied Genomics through the use of the Infinium HumanMethylation450 BeadChip Kit using manufacturer's protocols." The authors should here indicate how many subjects were evaluated and which tissue (cell type) was examined for methylation analysis.

(6) Methods. "M-value density and CpG-site intensity were plotted after normalization, and aberrant chips were removed." It would be helpful for the authors to state how many chips were removed and retained.

(7) Figure 1. Would it be possible in part A to indicate the location of cg05678571?

(8) Supplementary Figure 6. The first group shown in the plot (Homo Ref) contains only 2 observations. It doesn't really seem appropriate to show the box plot given this low sample size. It may be better to simply plot the two observations. Do the boxes span the IQR and do the whiskers denote $1.5 \times \text{IQR}$? This should be indicated in the legend.

(9) It may be helpful to include a supplemental table listing p-values obtained for top-ranked loci identified by previous genetic studies of otitis media (e.g., Allen et al. 2013, *J Assoc Res Otolaryngol* 14: 791-800; Rye et al. 2012, *PLoS One* 7, e48215). Apparently, such previously identified loci haven't reached genome-wide significance in the current analysis, but it would be helpful to see if these were of borderline significance or if the odds ratios are in the expected direction. This would help to consolidate the evolving information on the genetics of this disease and may suggest possible study-specific factors influencing GWAS outcomes.

Point-by-point responses to the reviewers comments of manuscript NCOMMS-16-02353

We appreciate the reviewers constructive suggestions and comments on our manuscript, and have addressed the points raised by the reviewers and elaborated more thoroughly in our point-by-point responses as outlined below.

Reviewer #1:

The article reports on FNDC1 as a novel GWAS locus for acute otitis media (AOM), which would be the first locus associated with AOM based on genome-wide statistical significance. The study results are moderately supported by established GWAS methods, including replication and meta-analysis, as well as methylation profile. The results are of particular interest to those who are involved in the genetics of otitis media, for which the previously identified association loci for common variants did not reach the genome-wide statistical significance threshold. There is some concern that there are major differences in phenotyping of each cohort, although if these phenotypic differences result in misclassification bias the expected effect on the study would be to diminish the OR, which might provide some relief that the association signal is robust to misclassification.

Response: Acute otitis media is a phenotype that can be defined by certain clinical parameters. The combination of these parameters accounts for the diagnosis in a subject who is then labeled with the corresponding ICD-9 code. The Generation R Study is a prospective cohort study with extensive data on clinical parameters, enabling researchers to discover the mildest of cases, however with less power compared to CHOP. CHOP is a large case-control study with extensive data on medical history through ICD-9 codes and with a strong case definition based on diagnoses made by medical professionals. We feel that between the studies, possible variability in phenotype definition is limited mostly to severity of disease, and is outweighed and resolved by the benefit of increased power. This is underlined by combined results showing similar direction of effect, similar magnitude of odds ratio. Furthermore, the P-value of homogeneity test in our meta-analysis is > 0.9 for the genome-wide significant SNP rs2932989, which indicates that there is no suggestion of between-study heterogeneity. We have included this information in the revised manuscript. We agree with the reviewer that considering the effect of possible misclassification bias, the signal appears to be robust. Skewness in case/control rate between both studies in Supplementary Table 2 is explained by differences in study design. As such, incidence rate of AOM can only be calculated from Generation R data, yet this is beyond the focus of this study.

What is the p-value for comparison of methylation profiles by additive genotype at rs2932989 (Fig. 1B)?

Response: We indicate in the manuscript that there is “significant association between methylation probe cg05678571 and rs2932989 (beta=-0.200, P=1.43x10⁻⁶, Fig. 1b)”

The authors themselves note that the lack of evidence of FNDC1 expression within the middle ear mucosa or the Eustachian tube is a limitation. Would it be possible for the authors to quickly test middle ear mucosa from a rodent model for FNDC1 expression?

Response: In an attempt to lend further support to the finding, we found that in additional 3 studies, Fndc1 expression was detected in mice ears. In addition, multiple cytokines which either regulate FNDC1 expression or function downstream of FNDC1 or in both ways, such as TNF- α , IL-1, have been shown to have middle ear expression and play important roles in otitis media.

Most importantly, in our extended experiment, we detected not only the expression of Fndc1 in C57/BL6 mice middle ear tissue (dominantly mucosal lining) by Western blotting, but also its upregulation upon LPS treatment compared to controls with PBS injection alone. The results have been summarized in new Supplementary Fig. 8. The description of the experiments and discussion of the results have been added into the maintext. Our results provide further supporting evidence for the involvement of FNDC1 in the etiology of OM.

Revised text: “In addition, Fndc1 expression was also found in pharyngeal tissues including the oral epithelium, next to esophagus tissue, lips, palate and salivary glands (Supplementary Table 7). Database “Bgee: Gene Expression Evolution” (<http://bgee.unil.ch/>) included three studies detecting the expression of Fndc1 in mice ears and one study showed its expression in inner ears; additional data from the Shared Harvard Inner-Ear Laboratory database (<https://shield.hms.harvard.edu>) indicates Fndc1 expression is present in both the cochlea and utricle. We further conducted experiments to examine the expression of Fndc1 in mouse middle ear tissue. By Western blotting, we observed not only the expression of Fndc1 in C57/BL6 mice middle ear tissue but also the upregulation of Fndc1 upon LPS treatment compared to control treated with PBS alone (Supplementary Fig. 8).”

It should be noted that FNDC1 is quite close to the PLG gene, and plg-deficient mice develop chronic otitis media (Eriksson et al. 2006). Is there any evidence that FNDC1 is related to other candidate genes for otitis media by gene ontology?

Response: The *PLG* gene is outside of the LD block (Figure 1) and topologically associating domains (TAD) (Supplementary Figure 10) where SNP rs2932989 is located at; *FNDC1* is the only gene in the same LD block and TAD with rs2932989. We tried to check the correlation between the most strongly associated SNP rs2932989 and *PLG* gene expression, using GTEx online portal. The *PLG* gene is “not sufficiently expressed” in almost all tissues in GTEx dataset, except testis, lung, whole blood and liver, where *PLG* expression did not show any correlation with rs2932989 genotype (P-value > 0.2). Thus, together with other supporting evidence we discussed in the manuscript, *FNDC1* is more likely to be the gene underlying this association.

In conjunction with Q9 from reviewer 2, we interrogated our GWAS dataset for previously proposed otitis media candidate genes, including genes from human genetic studies, gene expression and cellular function studies and/or model organism studies. We found 45 out of 82 candidate genes demonstrated evidence of nominally significant association, with intragenic SNPs or nearby SNPs of P-value < 0.05; Supplementary Table 6), such as *SMAD2*, *SMAD4*, *NELL1*, *BMP5*, *GALNT13* which are mediators of TGF-beta signaling, pro-inflammatory cytokine *IL1B* and its inhibitor *IL1RN*, receptor for fibroblast growth factor (*FGFR1*). TGF-beta, IL-1, FGF all have been reported to regulate *FNDC1* expression. Based on these analyses, *FNDC1* is related to multiple nominally significant OM candidate genes in several signaling pathways.

The authors should also include a list of suggestive loci identified from this study (if any) as these may also be of interest to other researchers.

Response: We have now added a new Supplementary Figure 6 and Supplementary Table 5, showing the regional association plot and the summary statistics of the index SNP at each locus with suggestive evidence of association ($5 \times 10^{-8} < \text{P-value} < 5 \times 10^{-5}$). Interestingly, several suggestive loci have been reported to be involved in pathogenesis of autoimmune disease, inflammatory disease or response to bacterial infection.

Revised text: “In addition to the genome-wide significant association at *FNDC1* locus, we also observed suggestive evidence for association at eight additional loci ($5 \times 10^{-8} < \text{P-value} < 5 \times 10^{-5}$; Supplementary Fig. 6 and Supplementary Table 5), among which, *KIF21B*, *CACNA1S*, *CRHR2*, *BDKRB2* and *TPM4* have been reported to be involved in pathogenesis of autoimmune disease, inflammatory disease or response to bacterial infection.”

Reviewer #2

This paper by van Ingen et al. reports findings from a replicated GWAS of acute otitis media. The authors identify one locus (6q25.3) with genome wide significance and show that potentially causal SNPs are associated with eQTL and methylation status. This is a high-quality study performed using sound and appropriate statistical analyses. I have made several suggestions for the authors that may help clarify certain aspects of the methodology (points 1 - 8 below).

(1) Results. "We found significant association between methylation probe cg05678571 and rs2932989 (beta=0.200, P=1.43x10⁻⁶, Fig. 1b) and even a stronger association between cg05678571 and SNP rs2501175 (beta=0.31, P=7.27x10⁻¹⁵; rs2501175 meta-analysis P=2.9x10⁻⁰⁷". It would be helpful here in the Results section to indicate the genome distance between cg05678571 and rs2932989, as well as the distance between cg05678571 and rs2501175.

Response: The genomic location of these 2 GWAS SNPs are: rs2932989 at chr6:159,699,284-159,699,284; rs2501175 at chr6:159,714,412-159,714,412; and the location of the methylation probe cg05678571 is at chr6:159,660,813-159,660,813.

The genomic distance between cg05678571 and rs2932989 is 38.5kb and that between cg05678571 and rs2501175 is 53.6kb, both of which are way under 1Mb, the average size of topologically associating domains (TAD) and gene co-expression clusters in human. Checking the 3D Genome Browser (<http://www.3dgenome.org>), we found that the above two GWAS SNPs and the methylation probe are indeed located in the same TAD (Supplementary Fig. 11), suggesting plausible interactions between these loci.

We have added the above information into our revised manuscript.

(2) Results. The authors list a number of tissues in which FNDC1 is expressed, including oral epithelium, esophagus tissue, lips, palate, salivary glands, inner ear, cochlea and utricle. The databases referenced by the authors, and other databases, however, include a number of tissue types; it would therefore be helpful to provide a ranked list of tissues and cell types based on FNDC1 expression (highest to lowest FNDC1 expression). For instance, a brief look at GTEx seemed to suggest that FNDC1 expression is highest in thyroid, sigmoid colon, and tibial nerve (not mentioned in the Results section). The BioGPS database indicates that FNDC1 expression is highest in mesenchymal stem cells, synovial membrane and fetal cartilage. These tissue types are not mentioned in the Results section.

Response: According to reviewer's suggestion, we have revised our manuscript including information on the expression level of FNDC1 in different human and mouse tissue/cell types. We showed the ranking of human FNDC1 expression level extracted from GTEx database in new Supplementary Figure 7. We also added Supplementary Data 1, showing rankings of human and mouse FNDC1 expression level extracted from databases BioGPS and Bgee. Data from Bgee database suggest that Fndc1 expression level in mouse ears is ranked 32 out of 217 mouse

tissue types. In our experiment, we detected the *Fndc1* protein in mice middle ears by western blotting and further showed its upregulation upon LPS treatment.

The revised text: “Expression data from both mouse and human tissues (<http://www.informatics.jax.org>; <http://www.eurexpress.org>; GTEx database) indicate *FNDC1* is expressed in a variety of different tissue types. Tissue types with top ranking expression levels of *FNDC1* were reported to be mesenchymal stem cells, synovial membrane, fetal cartilage, thyroid gland, adipose tissue and others (Supplementary Fig. 7; Supplementary Data 1). In addition, *FNDC1* expression was also found in pharyngeal tissues including the oral epithelium, next to esophagus tissue, lips, palate and salivary glands (Supplementary Table 7). Database “Bgee: Gene Expression Evolution” (<http://bgee.unil.ch/>) showed that three studies detected the expression of *Fndc1* in mice ears and one study showed its expression in inner ears; additional data from the Shared Harvard Inner-Ear Laboratory database (<https://shield.hms.harvard.edu>) indicates *Fndc1* expression is present in both the cochlea and utricle. We further conducted experiments to examine the expression of *Fndc1* in mouse middle ear tissue. By Western blotting, we observed not only the expression of *Fndc1* in C57/BL6 mice middle ear tissue but also the upregulation of *Fndc1* upon LPS treatment compared to control treated with PBS alone (Supplementary Fig. 8).”

***(3) Results. "In addition to the expression of FNDC1 in tissues of relevance to the phenotype, data from the GTEx database¹⁶ also indicates the AOM associated SNP, rs2932989, is an eQTL for FNDC1 in esophageal smooth muscle (Supplementary Fig. 6) ($P=1.9 \times 10^{-4}$)."* The authors should indicate whether this p-value reaches the genome-wide significance threshold, given the number of SNP-eQTL pairs evaluated.**

Response: GTEx now includes more samples in the updated version 6. We did the corresponding update in our new Supplementary Figure 9 (original Supplementary Figure 6) and updated the new P-value of 9.3×10^{-5} . It is not genome-wide significant, but it is “experiment-wide significant” considering the total number of 52 tissue types that we checked for SNP rs2932989.

GWAS SNPs and the expression of their underlying causal genes may not always show strong correlation. For example, Smemo et al. demonstrated that obesity-associated FTO intronic region has direct long range interaction with gene *IRX3* promoters and *IRX3* is the functional gene underlying the observed GWAS association. The authors showed the BMI GWAS SNP rs9930506 genotype is associated with *IRX3* gene expression in brain cerebellum at the significance level of $P=0.016$, which is only nominally significant. (Smemo S. et al., Nature, 2014). In addition, though GTEx is a good large reservoir for eQTL relationship, it did not show correlation of SNP gene expression in population of each specific ethnicity and did not have data specifically for the middle ear either. Thus we think though the correlation here is not genome-wide significant, it is still intriguing and supportive for *FNDC1* being the causal gene underlying the genome-wide significant association observed.

References:

Smemo S, et al. Obesity-associated variants within FTO form long-range functional connections with IRX3. *Nature* 507, 371-375 (2014)

(4) Results. "Two of the SNPs were found to have CTCF and EGR1 bound based on the ENCODE data19 (Supplementary Table 6)." Is the alignment of CTCF and EGR1 binding such that the sites are disrupted by genetic variation at the SNPs? It would be helpful here to indicate the cell types used for the ChIP assays.

Response: Using Haploreg search, we found that the CTCF conservative binding site was affected by the genetic variation of SNP rs1394234; there has been no report that SNP rs1553482 affects the conservative binding site of EGR1; however, ENCODE data showed the binding of proteins CTCF and EGR1 to these 2 sites respectively in the ChIP-Seq experiments, suggesting that the bindings sites could be conservative or non-conservative binding sites, or there may be the involvement of other transcription factors mediating the binding. In our original submission, we indicated the ID of the cell lines used in the ChIP assays; we have now added additional explanation of the cell types in the footnote of the new Supplementary Table 8 (original Supplementary Table 6). The cell types are not middle ear cell lines (There is no middle ear cell line in ENCODE study), however, similar underlying molecular mechanism may account for similar immune responses in these cell types. It would be interesting to carry out further experiments to test the binding of candidate transcription factors to specific sites in *FNDC1* in middle ear specific cell lines, but it is the beyond the scope of this manuscript.

(5) Methods. "Methylation status was ascertained for a subset of subjects recruited by the Center for Applied Genomics through the use of the Infinium HumanMethylation450 BeadChip Kit using manufacturer's protocols." The authors should here indicate how many subjects were evaluated and which tissue (cell type) was examined for methylation analysis.

Response: Genomic DNA was isolated from peripheral blood mononuclear cells. A total of 854 subjects were assayed on the Infinium HumanMethylation450 BeadChip. We have added such information into the paragraph of "Analysis of Methylation Data" in the methods section.

(6) Methods. "M-value density and CpG-site intensity were plotted after normalization, and aberrant chips were removed." It would be helpful for the authors to state how many chips were removed and retained.

Response: Only 2 chips were removed as aberrant, leaving 852 chips. Principle component analysis identified 425 Caucasians, 374 African Americans, 20 East Asians, and 24 Hispanics among these subjects. M-values (the log₂ ratio between the methylated and unmethylated probe intensities) were extracted and stored as a matrix. Additive genotypes at rs2932989 for subjects of European ancestry were extracted from existing genotyping data using PLINK. There are a total of 402 subjects of European ancestry without missing genotype at rs2932989 and extreme outlier values of methylation M-values (\geq median M-value of the genotype group \pm 3SD). Methylation data in gene *FNDC1* were analyzed as the response variable in a linear regression, with genotype at rs2932989 as the predictor variable among these 402 subjects. Sex, age, and 10 genotype-derived principle components were included as covariates. We have revised the paragraph of “Analysis of methylation data” in the methods section to incorporate such detailed information.

(7) Figure 1. Would it be possible in part A to indicate the location of cg05678571?

Response: We revised Figure 1, showing the position of cg05678571. In conjunction with reviewer’s Question 1, in the new Supplementary Figure 11 of the topologically associating domain at *FNDC1* locus, we also indicated the genomic positions of GWAS SNPs rs2932989 and rs2501175, as well as that of methylation probe cg05678571.

(8) Supplementary Figure 6. The first group shown in the plot (Homo Ref) contains only 2 observations. It doesn't really seem appropriate to show the box plot given this low sample size. It may be better to simply plot the two observations. Do the boxes span the IQR and to the whiskers denote 1.5*IQR? This should be indicated in the legend.

Response: The plot shown in the original Supplementary Figure 6 (Supplementary Figure 9 in the revised manuscript) is a screenshot from GTEx online portal. We don’t have enough individual level data and details to reproduce a dot plot. However, the individual data point for each sample has been shown on the box plot as grey circles.

The boxes span the IQR and to the whiskers denote 1.5*IQR. We also added the following explanation of the figure to the figure legend.

Revised figure legend: “The median expression level is shown by the black line in the box plot. The first quartile and the third quartile of the rank normalized gene expression are represented by the bottom and the top border of each box, respectively. The lowest and the highest datum within the 1.5 interquartil range (IQR) of the lower quartile and the upper quartile are shown by

the end of the lower whisker and that of the upper whisker, respectively. The data point of each individual is represented by the small grey circle.”

(9) It may be helpful to include a supplemental table listing p-values obtained for top-ranked loci identified by previous genetic studies of otitis media (e.g., Allen et al. 2013, J Assoc Res Otolaryngol 14: 791-800; Rye et al. 2012, PLoS One 7, e48215). Apparently, such previously identified loci haven't reached genome-wide significance in the current analysis, but it would be helpful to see if these were of borderline significance or if the odds ratios are in the expected direction. This would help to consolidate the evolving information on the genetics of this disease and may suggest possible study-specific factors influencing GWAS outcomes.

Response: According to reviewer’s suggestion, we added new Supplementary Table 6, listing the top SNP with the best P-value within or close to each OM candidate gene, including those from candidate gene studies (human genetics studies, cellular function studies and model organism studies) and the two GWAS studies mentioned by the reviewer. We indicated in the revised maintext that “Interrogating our GWAS dataset for candidate genes that have been previously proposed to be involved in pathogenesis of otitis media, we found 45 out of 82 genes demonstrated evidence of nominally significant association, with intragenic SNPs or nearby SNPs of P-value < 0.05; Supplementary Table 6), such as SMAD2, SMAD4, NELL1, BMP5, GALNT13 which are mediators of TGF-beta signaling, pro-inflammatory cytokine IL1B and its inhibitor IL1RN, receptor for fibroblast growth factor (FGFR1). TGF-beta, IL-1, FGF all have been reported to regulate FNDC1 expression.” Fourteen of these candidate genes were top ranked loci reported by previous GWASs, among which 9 showed at least nominally significant association in our study.

We also further address in the discussion section that “Among the genes that have been previously proposed to be involved in OM pathogenesis, we found more than half of them exhibited nominally significant association, suggesting consistency between genetic studies and/or with results from model organism studies. It is not surprising that some of the candidate genes did not show significant association, considering the following possible reasons. In our GWAS, we focused on a more defined phenotype of earl-onset AOM, which is different from the phenotypes examined in many of the previous OM genetic studies including OM in general, chronic OM or OM with effusion. In addition, different ethnicity of the study population is another influential factor to consider. Furthermore, polymorphisms of candidate genes, proposed based on evidence from differential gene expression, molecular, cellular functions and rodent model studies, may not always present an effect large enough to be captured by GWAS”.

Reviewers' Comments:

Reviewer #1 (Remarks to the Author)

I believe the authors have sufficiently addressed the reviewers' comments and questions. Save for a few minor errors that can be cleaned up during proofreading, the manuscript deserves publication and will be of interest to researchers working on otitis media and/or genetics.

Reviewer #2 (Remarks to the Author)

The authors have adequately addressed my comments and I find the revised manuscript to be improved. I have no further comments to provide at this time.

Point-by-point responses to the reviewers comments of manuscript NCOMMS-16-02353A.

Reviewer #1

I believe the authors have sufficiently addressed the reviewers' comments and questions. Save for a few minor errors that can be cleaned up during proofreading, the manuscript deserves publication and will be of interest to researchers working on otitis media and/or genetics.

Response: We appreciate the reviewer's comment on our revision. The reviewer's comments and recommendations have improved our manuscript. We have also gone through the manuscript carefully and corrected minor errors as shown by track changes.

Reviewer #2

The authors have adequately addressed my comments and I find the revised manuscript to be improved. I have no further comments to provide at this time.

Response: We appreciate the reviewer's comment on our revision. The reviewer's comments have improved the manuscript and we are pleased that our revision meets the reviewer's approval.